# Automated Machine Learning in Predicting 30-Day Mortality in Patients with Non-Cholestatic Cirrhosis

**DOI:** 10.3390/jpm12111930

**Published:** 2022-11-19

**Authors:** Chenyan Yu, Yao Li, Minyue Yin, Jingwen Gao, Liting Xi, Jiaxi Lin, Lu Liu, Huixian Zhang, Airong Wu, Chunfang Xu, Xiaolin Liu, Yue Wang, Jinzhou Zhu

**Affiliations:** 1Department of Gastroenterology, The First Affiliated Hospital of Soochow University, 188 Shizi Street, Suzhou 215006, China; 2Suzhou Clinical Center of Digestive Diseases, Suzhou 215000, China; 3Department of Gastroenterology, Dushu Lake Hospital Affiliated to Soochow University, Suzhou 215000, China; 4Department of Hepatology, The Fifth People’s Hospital of Suzhou, Suzhou 215000, China

**Keywords:** non-cholestatic cirrhosis, automated machine learning, shapley additive explanation, partial dependence plots, local interpretable model agnostic explanation

## Abstract

Objective: To evaluate the feasibility of automated machine learning (AutoML) in predicting 30-day mortality in non-cholestatic cirrhosis. Methods: A total of 932 cirrhotic patients were included from the First Affiliated Hospital of Soochow University between 2014 and 2020. Participants were divided into training and validation datasets at a ratio of 8.5:1.5. Models were developed on the H_2_O AutoML platform in the training dataset, and then were evaluated in the validation dataset by area under receiver operating characteristic curves (AUC). The best AutoML model was interpreted by SHapley Additive exPlanation (SHAP) Plot, Partial Dependence Plots (PDP), and Local Interpretable Model Agnostic Explanation (LIME). Results: The model, based on the extreme gradient boosting (XGBoost) algorithm, performed better (AUC 0.888) than the other AutoML models (logistic regression 0.673, gradient boost machine 0.886, random forest 0.866, deep learning 0.830, stacking 0.850), as well as the existing scorings (the model of end-stage liver disease [MELD] score 0.778, MELD-Na score 0.782, and albumin-bilirubin [ALBI] score 0.662). The most key variable in the XGBoost model was high-density lipoprotein cholesterol, followed by creatinine, white blood cell count, international normalized ratio, etc. Conclusion: The AutoML model based on the XGBoost algorithm presented better performance than the existing scoring systems for predicting 30-day mortality in patients with non-cholestatic cirrhosis. It shows the promise of AutoML in its future medical application.

## 1. Introduction

Cirrhosis is a leading cause of morbidity and mortality across the world, which is characterized by a systemic pro-inflammatory milieu consisting of persistent liver inflammation, extracellular matrix remodeling, and the accumulation of collagen in liver tissue [1]. Etiologically, viral hepatitis, especially hepatitis B viral (HBV), has been the leading pathogeny of cirrhosis. In addition, several kinds of cirrhosis were associated with cholestasis, such as primary sclerosing cholangitis (PSC) and primary biliary cholangitis (PBC) [2]. Complications mainly include ascites, portal hypertensive gastrointestinal bleeding, jaundice, coagulopathy, and hepatic encephalopathy which may recur with increasing frequency after the initial presentation, and most patients die within a median time of approximately 2 years [3,4]. Most studies support that early intervention in stabilized cirrhosis may delay its progression to the decompensated stage [5].

Several prediction models for the prognosis of liver cirrhosis have been proposed. The Child-Turcotte-Pugh score (CTP), including serum levels of bilirubin and albumin, prothrombin time, degree of ascites, and severity of hepatic encephalopathy, was initially introduced to predict mortality in patients with cirrhosis who underwent surgery [6]. It consists of two subjective indicators, which were prone to bias during application. The model for end-stage liver disease (MELD) score was validated as a predictor of allocation of organs for liver transplantation, rather than originally used to assess mortality in patients undergoing trans-jugular intrahepatic portosystemic shunt (TIPS) [7,8]. Even though these above models were classical, they ignored the effect of cholestasis on lipid metabolism in patients with cirrhosis.

Machine learning (ML) is a scientific discipline that emphasizes efficient computing algorithms, including supervised, unsupervised, semi-supervised, and reinforcement learning, which are widely used in medicine [9]. Traditional ML includes support vector machine (SVM), gradient boosting machine (GBM), extreme gradient boosting (XGBoost), etc. Automated machine learning (AutoML) intelligently computes hundreds or even thousands of mathematical models and eventually filters out the optimal model, which is more suitable for clinicians without a computer foundation.

A series of previous studies has proven that clinical models based on machine learning performed better than models based on traditional logistic regression. There were no previous reports concerning AutoML and cirrhosis, thus we conducted this hospital-based case-control study to develop AutoML models for predicting 30-day mortality in patients with non-cholestatic cirrhosis. On one hand, we evaluate the feasibility of AutoML in the management of chronic liver disease. On the other hand, we observe the performance of AutoML in clinical modeling based on electronic health records. In this study, multiple models, based on various AutoML algorithms, were developed and compared with the existing scoring systems.

## 2. Materials and Methods

### 2.1. Study Design

This was a hospital-based cohort study involving 932 patients with non-cholestatic cirrhosis from the First Affiliated Hospital of Soochow University between 2014 and 2020. A random grouping method was used to divide all participants into the training and validation datasets according to the ratio of 8.5:1.5, consisting of 792 and 140 patients, respectively. The outcome was 30-day mortality since hospitalization. All participants had signed informed consent for the medical data used in the study. According to the principles of the Declaration of Helsinki, the study was approved by local Institutional Review Boards.

A series of AutoML algorithms on the H_2_O platform were used for modelling in the training dataset. H_2_O, an open source and scalable platform for machine learning, provided multiple algorithms and used K-fold cross-validation to develop and validate models [10]. Models were evaluated by area under receiver operating characteristic (ROC) curves (AUC) and compared with the existing scorings, including the model of end-stage liver disease (MELD) score, the MELD-Na score, and the albumin-bilirubin (ALBI) score, in the training and validation datasets. The flowchart is plotted in Figure 1.

### 2.2. Inclusion and Exclusion Criteria

Subjects included had clear evidence of non-cholestatic liver cirrhosis either by liver histology or by clinical presentations, liver function tests, and medical imaging techniques. Exclusion criteria included the presence of solid organ transplantation, malignant neoplasm, PBC, and lack of complete medical records. The diagnosis of PBC is based on the following criteria [2]: (1) biochemical evidence of cholestasis with an elevation of alkaline phosphatase activity; (2) presence of antimitochondrial antibody (AMA); (3) histopathologic evidence of nonsuppurative cholangitis and destruction of small or medium-sized bile ducts if a biopsy is performed.

### 2.3. Clinical Data Collection

The data of all participants were retrospectively enrolled from the Department of Gastroenterology and Hepatology, The First Affiliated Hospital of Soochow University. Demographic information included gender, age, body mass index (BMI), history of hypertension, history of diabetes, etiology of cirrhosis, and complications of cirrhosis. Laboratory parameters include white blood cell count (WBC), platelet count (PLT), total bilirubin (TBIL), creatinine, alanine aminotransferase (ALT), aspartate aminotransferase (AST), albumin, sodium, total cholesterol (TC), triglycerides (TG), high-density lipoprotein cholesterol (HDL-C), low-density lipoprotein cholesterol (LDL-C), prealbumin, prothrombin time (PT), and international standard ratio (INR). Biochemical analyses were all available from a Hitachi 7600 Auto-Analyzer (Hitachi, Tokyo, Japan) or an Abbott-Architect Immunoanalyzer (Abbott Laboratories, Abbott Park, IL, USA). According to survival conditions 30 days after admission, participants were separately divided into two groups, namely the death and survival groups. The calculation methods of MELD score [7], MELD-Na score [11], and ALBI score [12] were as follows:

MELD = 3.78 × ln[serum bilirubin (mg/dL)] + 11.2 × ln[INR] + 9.57 × ln[serum creatinine mg/dL] + 6.43.

MELD-Na = MELD-Na − [0.025 × MELD × (140 − Na)] + 140, where the serum sodium concentration is bound between 125 and 140 mmol/L.

ALBI = 0.66 × log[serum bilirubin (mg/dL)] − 0.085 × protein (mg/dL).

### 2.4. Statistics Analysis

Data analysis was performed by R (version 4.1.0, R Foundation for Statistical Computing, Vienna, Austria), and SPSS (version 24.0, SPSS Inc., Chicago, IL, USA). Continuous variables were expressed as median (Q1–Q3) or mean ± standard, and categorical variables were expressed by frequencies and percentages. Binary logistic regression analysis and automated machine learning based on the H_2_O package were used to establish the predictive model for non-cholestatic cirrhosis. The discrimination of the model was assessed by using receiver operating curves (ROC). Our study utilized three methods for model interpretation: the SHapley Additive exPlanations (SHAP) analysis, Partial Dependency Plots (PDP), and Local Interpretable Model Agnostic Explanation (LIME), which represented how the features affect the output of the model’s prediction. A two-sided *p* < 0.05 was considered statistically significant.

## 3. Results

### 3.1. Characteristics of the Patients

A total of 932 cirrhotic patients were enrolled in this study, including 136 in the non-survival group and 796 in the survival group. The 30-day mortality rate was 14.6%. Further details on the characteristics of patients for 30-day mortality are given in Table 1.

### 3.2. Models Based on AutoML Algorithms

In the validation dataset, the AutoML model based on the XGBoost algorithm showed the best performance in discrimination (AUC = 0.888) when compared to logistic regression (AUC = 0.673), gradient boost machine (AUC = 0.886), random forest (AUC = 0.866), deep learning (AUC = 0.830), stacking (AUC = 0.850), MELD (AUC = 0.778), MELD-Na (AUC = 0.782), and ALBI (AUC = 0.662) as shown in Table 2.

### 3.3. Interpretation of the AutoML Model Based on XGBoost Algorithm

Figure 2 shows the ten key variables in the AutoML model based on the XGBoost algorithm. HDL-C was the foremost feature, followed by WBC, creatinine, age, PT, TC, albumin, etc.

Moreover, we plotted SHAP, which is a unified approach for explaining the outcome of any machine learning mode, to provide consistent and accurate attribution values for each feature. Figure 3 presented that the closer the values of features in the model were to 1.5, the closer the correlations were with 30-day mortality of cirrhosis.

PDP were drawn to more clearly show the relationship between various variables and mortality of cirrhosis (Figure 4). Among the key variables, creatinine, WBC, INR, and age showed upward trends with the mortality, while HDL-C and TC showed opposite trends.

As shown in Figure 5, LIME plot shows how the important variables (in the XGBoost model) contributed to the mortality based on four selected samples. For example, case #4 had a high probability of 0.68 for death due to cirrhosis as predicted by the XGBoost model. HDL-C, creatinine, TC, WBC, and age played distinct roles in the prediction.

## 4. Discussion

Based on the clinical data of the 932 cirrhotic patients, we have developed and validated a series of AutoML models in predicting 30-day mortality.

We found the AutoML model based on the XGBoost algorithm showed best performance of discrimination when compared to the other models and the existing scoring systems. Furthermore, we visualized the best model for interpretation using SHAP, PDP, and LIME.

In terms of the key variables for the mortality in cirrhosis, we found HDL-C was the most important variable in the XGBoost model. The level of HDL-C was associated with the severity of end-stage liver disease [13,14]. Terib et al. demonstrated that HDL-C level was a robust predictor for survival in patients with chronic liver failure [15]. Habib et al. suggested that HDL-C was significantly decreased in non-cholestatic cirrhotic veterans [16]. The mechanisms underlying the decrease in HDL-C level due to cirrhosis are unclear, but recent studies have focused on the anti-inflammatory effects of HDL-C. HDL-C can decrease cholesterol levels in cell membranes by removing intracellular lipids, thereby reducing lipid secretion of pro-inflammatory cytokines [17]. According to the published studies, we have observed that cirrhosis impaired the ability of HDL-C to inhibit lipopolysaccharide-induced (LPS) activation of the pro-inflammatory transcription factor NF-kB and subsequent production of interleukin-6 (IL-6), interleukin-8 (IL-8, and tumor necrosis factor-α (TNF-α) in monocytes [18]. In addition, a study in patients with advanced chronic liver failure showed that the addition of recombinant HDL-C to restore HDL function reduced the LPS-induced inflammatory response [19]. The evidence was confirmed in various animal models [20].

We also found that higher WBC counts at admission were associated with mortality, which was inconsistent with a previous study in the United States [21]. It is possible that cirrhotic patients who mount a leukocytosis have a higher pro-inflammatory milieu, resulting in a decrease in survival rate [22,23]. Moreover, consistent with the results of several previous studies [24,25], elevated creatinine levels were an associated risk factor for poor prognosis in cirrhosis, which is mainly related to vasodilatory mechanisms and non-vasodilatory mechanisms identified in recent years [26]. The vasodilatory mechanism is mainly related to the dilatation of visceral arterial vessels during the decompensated phase of cirrhosis, the reduction of systemic effective circulating blood volume, the reduction of cardiac output, the activation of the renin-angiotensin-aldosterone system, and the impaired renal function due to insufficient renal perfusion [27]. Lastly, various inflammatory factors, intra-abdominal hypertension, high bilirubin and bile acids, relative adrenal insufficiency, and other non-vasodilatory mechanisms also play an important role in creatinine elevation [28,29].

In addition, this study found that PT and INR values were larger in the mortality group than in the survival group, suggesting that coagulation function was a good predictive indicator of 30-day mortality in non-cholestatic cirrhosis. P. G. Northup et al. reported that the INR and PT were inextricably linked to the prognosis and progression of liver disease [30]. Both J. Li et al. [31] and T. Wu et al. [32] developed predictive models on the prognosis of chronic liver disease and also included INR, which was mainly related to the involvement of the liver in coagulation factors. This study also revealed that age was an independent risk factor for poor prognosis of cirrhosis, consistent with recent epidemiological findings [33], which may be associated with weakened immune systems in the elderly.

A good predictive model can effectively and accurately predict the prognosis of disease, and facilitate the early monitoring of high-risk patients to reduce the mortality of the disease in question. In the study, we reported the application of AutoML in predicting mortality of cirrhosis for the first time. We found the model based on the XGBoost algorithm showed the best performance of discrimination when compared to the existing scoring systems, e.g., MELD, MELD-Na, and ABLI. To overcome the drawback of the black box in machine learning, we interpreted the model using various means of visualization.

There are several limitations in this study: first, this is a single-central study that may lead to results bias and multi-central data are needed to validate the new model; second, our results are observational, which could not prove causality and may be influenced by unmeasured confounders; finally, the outcome is only designed for 30-day mortality and may require further long-term follow-up.

In this study, a series of AutoML models were developed for predicting 30-day mortality in patients with non-cholestatic cirrhosis. The model, based on the XGBoost algorithm, presented best performance when compared to existing scoring systems. It shows the promise of AutoML in its future medical application.

## Figures and Tables

**Figure 1 jpm-12-01930-f001:**
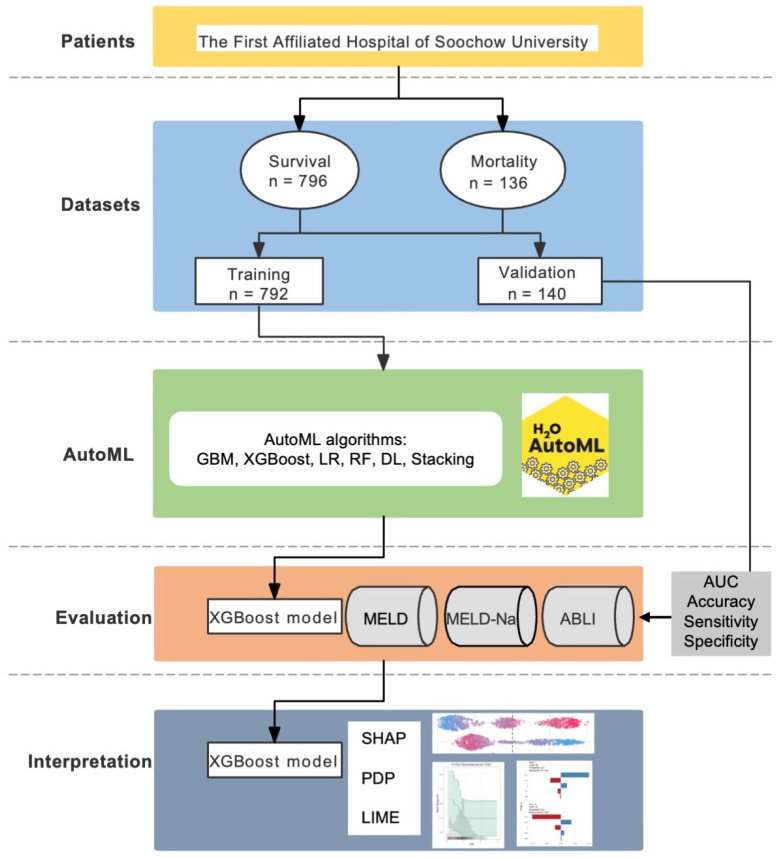
Flowchart of the study.

**Figure 2 jpm-12-01930-f002:**
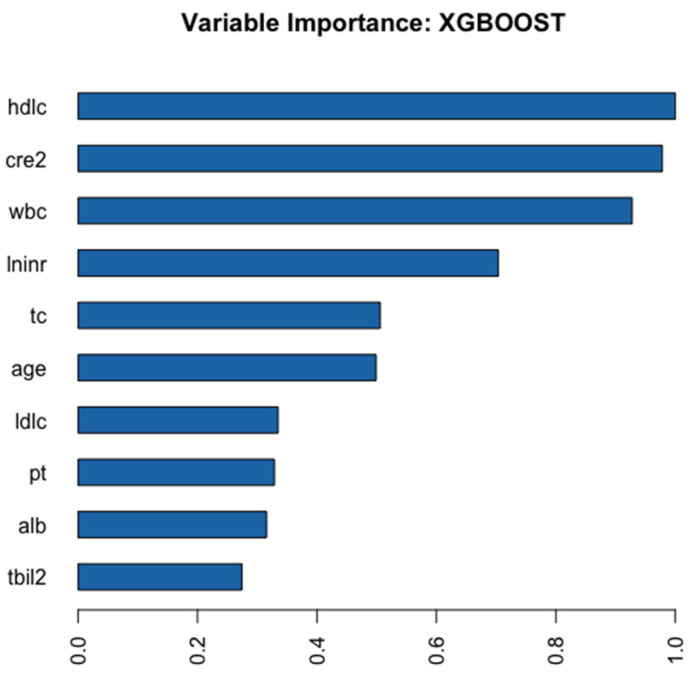
Relative importance of the variables in the XGBoost model. The chart shows that HDL-C was the most important variable, followed by Cre, WBC, INR, etc. Abbreviations: hdlc, high-density lipoprotein cholesterol; cre2, creatinine; wbc, white blood cell count; lninr, international normalized ratio; tc, total cholesterol; ldlc, low-density lipoprotein cholesterol; pt, prothrombin time; alb, albumin; tbil2, total bilirubin.

**Figure 3 jpm-12-01930-f003:**
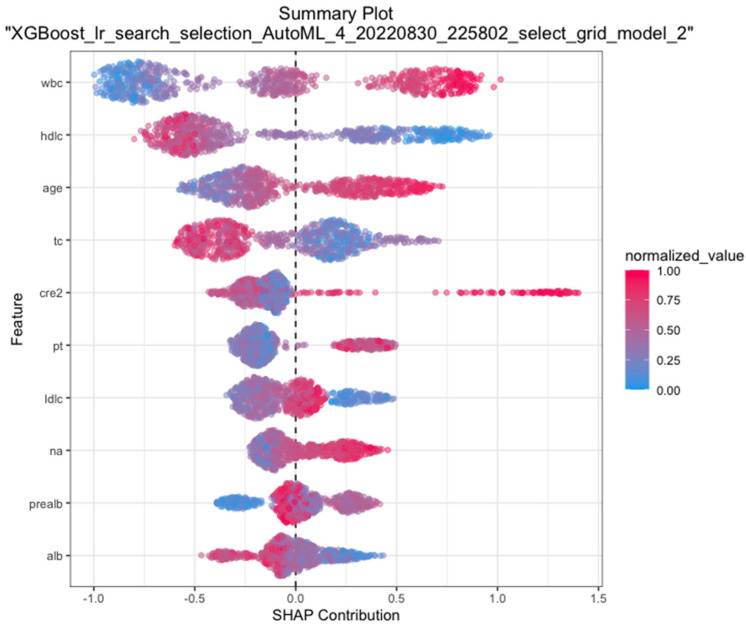
SHAP plotting of the XGBoost model. The closer the variable value is to 1.5 at the *x*-axis, the higher the possibility of mortality at 30-day. Abbreviations: wbc, white blood cell count; hdlc, high-density lipoprotein cholesterol; tc, total cholesterol; cre2, creatinine; pt, prothrombin time; ldlc, low-density lipoprotein cholesterol; na, blood sodium; pre-alb, pre-albumin; alb, albumin; SHAP, SHapley additive explanation.

**Figure 4 jpm-12-01930-f004:**
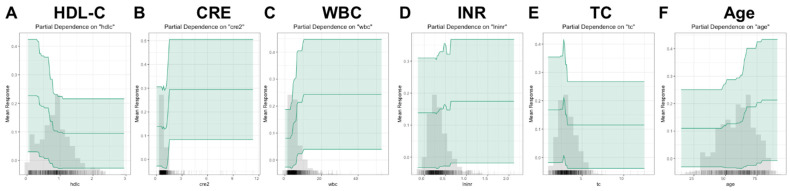
PDP for the key variables in the XGBoost model. The chart shows that creatinine, white blood cell, international normalized ratio, and age showed upward trends with the mortality, while high-density lipoprotein cholesterol and total cholesterol showed opposite trends. Abbreviations: PDP, partial dependence plot; HDL-C, high-density lipoprotein cholesterol; CRE, creatinine; WBC, white blood cell count, INR, international normalized ratio; TC, total cholesterol.

**Figure 5 jpm-12-01930-f005:**
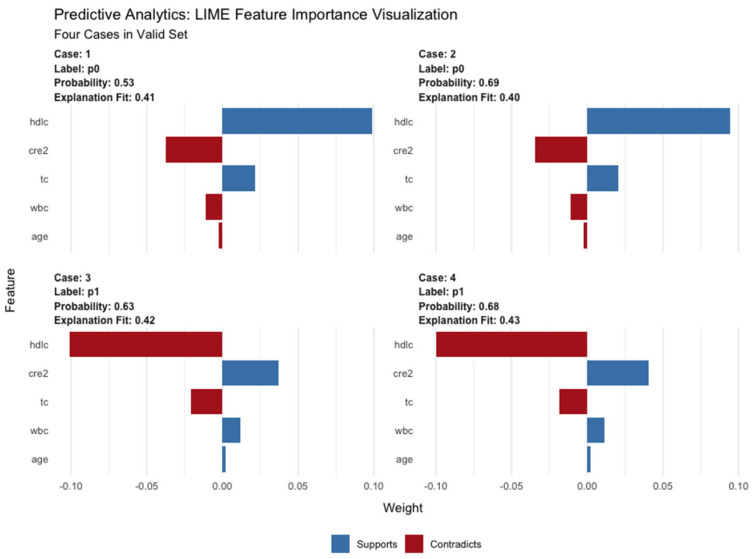
LIME of the XGBoost model in four cases from the validation set. The graph of LIME shows how the important variables (in the XGBoost model) contributed to mortality based on four selected samples (two from the survival group, two from the mortality group). Lable p0 means a survival sample, while lable p1 means a dead sample. Abbreviations: LIME, Local Interpretable Model Agnostic Explanation; hdlc, high-density lipoprotein cholesterol; cre2, creatinine; tc, total cholesterol; wbc, white blood cell count.

**Table 1 jpm-12-01930-t001:** Characteristics of patients with non-cholestatic cirrhosis by 30-day mortality.

	Total (n = 932)	Survival (n = 796)	Mortality (n = 136)	*p*
Sex, n (%)				0.742
Male	581(62)	494(62)	87(64)	
Female	351(38)	302(38)	49(36)	
Age (years)	61(50, 69)	60(49, 69)	67(53, 73)	<0.001
BMI	23.24 ± 3.19	23.33 ± 3.14	22.50 ± 3.56	0.074
Etiology				0.784
ALD	80(9)	67(8)	13(10)	
Others	852(91)	729(92)	123(90)	
Complication, n (%)				<0.001
Ascites	123(13)	115(14)	8(6)	
Hypersplenism	69(7)	67(8)	2(1)	
EGVB	283(30)	247(31)	36(26)	
HE	102(11)	79(10)	23(17)	
Hypohepatia	355(38)	288(37)	67(49)	
WBC (1012 g/L)	4.87(3.30, 7.23)	4.53(3.15, 6.48)	7.29(4.91, 10.91)	<0.001
PLT (109 g/L)	77.00(50.00, 124.25)	78.00(51.00, 126.00)	76.00(49.00, 112.25)	0.489
TBIL (μmol/L)	2.14(1.23, 4.46)	1.99(1.20, 3.75)	4.93(1.93, 13.59)	<0.001
Creatinine (mg/dL)	0.74(0.60, 0.94)	0.73(0.59, 0.92)	0.84(0.65, 1.40)	<0.001
ALT (U/L)	28.00(18.00, 52.02)	27.55(17.90, 48.00)	33.10(19.67, 89.50)	0.003
AST (U/L)	41.65(28.67, 71.75)	40.00(28.08, 67.53)	52.95(32.88, 106.40)	<0.001
Albumin (g/L)	30.40(26.67, 34.10)	30.70(27.00, 34.70)	28.00(23.98, 31.50)	<0.001
Na (mmol/L)	139.50(137.00, 141.80)	139.50(137.00, 141.80)	138.30(134.67, 141.27)	0.01
TC (mmol/L)	3.14(2.50, 3.91)	3.21(2.55, 4.00)	2.64(2.00, 3.17)	<0.001
TG (mmol/L)	0.88(0.66, 1.24)	0.88(0.66, 1.22)	0.94(0.64, 1.30)	0.4
HDL-C (mg/dL)	0.87(0.52, 1.17)	0.90(0.62, 1.21)	0.47(0.21, 0.78)	<0.001
LDL-C (mg/dL)	1.60(1.18, 2.16)	1.65(1.26, 2.20)	1.21(0.89, 1.85)	<0.001
Pre-ALB (mg/dL)	73.35(48.00, 104.65)	76.55(50.98, 108.20)	58.90(39.93, 83.30)	<0.001
PT (s)	15.50(13.70, 17.90)	15.20(13.50, 17.30)	18.30(15.60, 22.02)	<0.001
LnINR	0.30(0.17, 0.45)	0.28(0.17, 0.42)	0.48(0.30, 0.68)	<0.001
MELD	9.89(6.16, 14.83)	9.41(5.77, 13.37)	17.11(9.87, 23.73)	<0.001
MELD-Na	10.43(5.54, 16.46)	9.62(5.10, 14.94)	18.07(10.30, 25.39)	<0.001
ALBI	−1.53 ± 0.66	−1.59 ± 0.64	−1.14 ± 0.62	<0.001

Abbreviations: ALBI, albumin-bilirubin; ALD, alcoholic liver disease; ALT, alanine transaminase; AST, aspartate aminotransferase; EGVB, esophagogastric variceal bleeding; HDL-C, high-density lipoprotein cholesterol; HE, hepatic encephalopathy; LDL-C, low-density lipoprotein cholesterol; LnINR, Ln(international normalized ratio); MELD, a model for end-stage liver disease; MELD-Na, MELD combined serum sodium concentration; PLT, platelet count; Pre-ALB, prealbumin; PT, prothrombin time; TBIL, total bilirubin; TC, cholesterol; TG, triglyceride; WBC, white blood cell.

**Table 2 jpm-12-01930-t002:** Performance of the AutoML models and the existing scoring systems in predicting mortality in patients with non-cholestatic cirrhosis.

Dataset	Algorithm	AUC	Accuracy	Sensitivity	Specificity
Training	GBM	0.900	0.876	0.597	0.926
	XGBoost	0.938	0.915	0.647	0.963
	LR	0.662	0.852	0.555	0.905
	RF	0.807	0.833	0.513	0.890
	DL	1.000	0.999	1.000	0.999
	Stacking	0.956	0.922	0.655	0.969
	MELD	0.806	0.784	0.736	0.793
	MELD-Na	0.791	0.797	0.669	0.820
	ALBI	0.739	0.649	0.752	0.630
Validation	GBM	0.886	0.857	0.412	0.919
	XGBoost	0.888	0.879	0.471	0.935
	LR	0.673	0.821	0.353	0.886
	RF	0.866	0.821	0.294	0.894
	DL	0.830	0.850	0.471	0.902
	Stacking	0.850	0.871	0.294	0.951
	MELD	0.778	0.864	0.588	0.902
	MELD-Na	0.782	0.857	0.588	0.894
	ALBI	0.662	0.536	0.824	0.496

Abbreviations: DL, deep learning; GBM, gradient boost machine; LR, logistic regression; RF, random forest; XGBoost, eXtreme gradient boosting.

## Data Availability

Not applicable.

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
