# Peer review of "Automated Machine Learning in Predicting 30-Day Mortality in Patients with Non-Cholestatic Cirrhosis"

_jpm, 2022, doi:10.3390/jpm12111930_

Round 1
Reviewer 1 Report
I read with interest the paper written by Chenyan Yu et al.
Some points:
- Line 59: … they ignored the effect of cholestasis on lipid metabolism in patients with cirrhosis… It would be nice to have more insight into the role of cholestasis, why it should be prognostic, and should be included in a prognostic score
- Inclusion criteria: I think this is the main weakness of the article. What do you consider cholestatic? No doubt when you have liver biopsy available, but what is your labs threshold to consider the disease cholestatic? Not always high bilirubin consists of cholestasis and vice versa. How do you address this point?
- The paper is very technical with few explanations. Would be beneficial to have more explanation about the role of HDL-C and other markers comparing the current literature and defining their effect/role in the prognostic setting.
Author Response
- Line 59: … they ignored the effect of cholestasis on lipid metabolism in patients with cirrhosis… It would be nice to have more insight into the role of cholestasis, why it should be prognostic, and should be included in a prognostic score.
= Thanks a lot for your valuable comments. We do realize that it plays an important role of cholestasis in cirrhosis prognosis. In the meantime, patients with non-cholestatic cirrhosis and patients with cholestatic cirrhosis show remarkable heterogeneity. The liver is involved in the synthesis and metabolism of lipids, so cholestasis must affect the values of laboratory lipid tests. In this study, relevant variables such as triglycerides and serum cholesterol were included which could be affected by cholestatic cirrhosis. Thus, we fitted models in patients with non-cholestatic cirrhosis in this study.
- Inclusion criteria: I think this is the main weakness of the article. What do you consider cholestatic? No doubt when you have liver biopsy available, but what is your labs threshold to consider the disease cholestatic? Not always high bilirubin consists of cholestasis and vice versa. How do you address this point?
= Cholestasis is defined as a disorder of bile secretion or excretion due to multiple causes, so that bile cannot flow normally into the duodenum and even involves the liver, resulting in cholestatic liver disease. In this study, diagnostic criteria for cholestasis as follows: biochemical tests revealed serum alkaline phosphatase(ALP) greater than 1.5 times the upper limit of normal(ULN) and γ-glutamyl aminotransferase(γ-GT) greater than 3 times the ULN.
- The paper is very technical with few explanations. Would be beneficial to have more explanation about the role of HDL-C and other markers comparing the current literature and defining their effect/role in the prognostic setting.
= In addition, this study found that PT and INR values were larger in mortality group than in the survival group, suggesting that coagulation function was a good predictive indicator of 30-day mortality in non-cholestatic cirrhosis. P. G. Northup et al. reported that the INR and PT were inextricably linked to prognosis and progression of liver disease. Both J. Li et al. and T. Wu et al. developed predictive models on the prognosis of chronic liver disease and also included INR, which was mainly related to the involvement of the liver in coagulation factors. This study also revealed that age was an independent risk factor for poor prognosis of cirrhosis, consistent with recent epidemiological findings, which may be associated with weakened immune system in the elderly. We have provided a more detailed explanation of the variables. Please see the highlighted. (Line 237-245)
Reviewer 2 Report
Overall, the article is very well written and the conclusion is well supported by the results. I would suggest just minor spell check before publication.
Author Response
Overall, the article is very well written and the conclusion is well supported by the results. I would suggest just minor spell check before publication.
= Your advices are valuable to our work. The manuscript had been revised by an English professor. Please see the highlighted.
Reviewer 3 Report
This manuscript describes automated ML for mortality prediction in non-cholestasis cirrhosis. The manuscript was well prepared.
There are no points of concern in the central part of this manuscript.
Moreover, the cited and their discussion are presented in good style. However, authors have some minor points that needs to be addressed.
1. Man" and "Women" in Table 1 should be written as "male" and "female" in normal papers.
2. Additionally, the abbreviations listed below the table should be listed in the order in which they appear or in alphabetical order.
3. Figure 4 is too small to see.
Please use a larger and higher resolution image.
1. It is common to describe gender as male or female instead of man or woman.
Author Response
This manuscript describes automated ML for mortality prediction in non-cholestasis cirrhosis. The manuscript was well prepared.
There are no points of concern in the central part of this manuscript.
Moreover, the cited and their discussion are presented in good style. However, authors have some minor points that needs to be addressed.
- Man" and "Women" in Table 1 should be written as "male" and "female" in normal papers.
= Thanks a lot for your suggestion. Table 1 had been revised. Please see the highlighted.
- Additionally, the abbreviations listed below the table should be listed in the order in which they appear or in alphabetical order.
= Your advices are valuable to our work. The abbreviations listed below Table 1 and Figure 3 had been revised. Please see the highlighted.
- Figure 4 is too small to see. Please use a larger and higher resolution image.
= Figure 4 had been replaced with a larger and higher resolution image. Thank you for your valuable suggestion.
Reviewer 4 Report
In the manuscript entitled “Automated machine learning in predicting 30-day mortality in patients with non-cholestatic cirrhosis”, the study is merit. However, minor issues may be raised to improve the manuscript.
1- Some grammar and typos errors should be revised
2- The objectivity of the study should be discussed in more detail.
3- The conclusion section in the abstract should be corrected to be concise and targeted.
4- How to correlate your study with previous studies like doi: 10.1371/journal.pone.0256428? what is the new information that your manuscript introduces?
Author Response
In the manuscript entitled “Automated machine learning in predicting 30-day mortality in patients with non-cholestatic cirrhosis”, the study is merit. However, minor issues may be raised to improve the manuscript.
1-Some grammar and typos errors should be revised
= Your advices are valuable to our work. The manuscript had been revised by an English professor. Please see the highlighted.
2-The objectivity of the study should be discussed in more detail.
= A series of previous studies have proven that clinical models based on machine learning performed better than models based on traditional logistic regression. There were no previous reports concerned AutoML and cirrhosis, thus we conducted this hospital-based case-control study to develop AutoML models for predicting 30-day mortality in patients with cirrhosis. For one hand, we evaluate the feasibility of AutoML in the management of chronic liver disease. For the other hand, we observe the performance of AutoML in clinical modeling based on electronic health records. In this study, multiple models, based on various AutoML algorithms, were developed and compared with the existing scoring systems. We revised the end of the introduction, based on your suggestion. Please see the highlighted. (Line 68-76)
3-The conclusion section in the abstract should be corrected to be concise and targeted.
= The conclusion in the abstract had been revised based on your suggestion.
The AutoML model based on XGBoost algorithm presented better performance than the existing scorings for predicting 30-day mortality in patients with non-cholestatic cirrhosis. It shows the promise of AutoML in the future medical application.
Please see the highlighted. (Line 32-34)
4-How to correlate your study with previous studies like doi: 10.1371/journal.pone.0256428? what is the new information that your manuscript introduces?
= Thanks a lot for your review. Your inputs are valuable to our work. In the previous study, Guo et al. manually developed prediction models for mortality in patients with cirrhosis, based on machine learning algorithms. In our study, the models were trained on an automated machine learning platform (H2O.ai). Moreover, the outcome in our study was short-term death (30 days), while the outcomes in Gao et al. study were mid-/long-term mortality (90, 180, and 365-day). Last, Guo et al. fitted models using electronic health record data from a large academic liver transplant center in the west, in which 77.5% patients were white and 15.7% were black. However, all of the patients in our study were Chinese. The etiology and management of cirrhosis are different between the west and the east.
Round 2
Reviewer 1 Report
Dear authors,
Thank you for the reply. However, even if improved, I think the study does not have the requirement for publication. More insight and references on cholestatic/noncholestatic disease, how you set your inclusion criteria, and the pathogenesis are essential to understand the prognostic role of a new model score.